# Multispectral Imaging of Collagen, NAD(P)H and Flavin Autofluorescence in Mesenchymal Stem Cells Undergoing Trilineage Differentiation

**DOI:** 10.3390/cells13201731

**Published:** 2024-10-18

**Authors:** Jared M. Campbell, Saabah B. Mahbub, Ayad G. Anwer, Abbas Habibalahi, Stan Gronthos, Sharon Paton, Shane T. Grey, Lindsay E. Wu, Robert B. Gilchrist, Ewa M. Goldys

**Affiliations:** 1Graduate School of Biomedical Engineering, University of New South Wales, Sydney, NSW 2052, Australiaa.anwer@unsw.edu.au (A.G.A.); a.habibalahi@gmail.com (A.H.); e.goldys@unsw.edu.au (E.M.G.); 2Mesenchymal Stem Cell Laboratory, School of Biomedicine, Faculty of Health and Medical Sciences, University of Adelaide, Adelaide, SA 5000, Australia; stan.gronthos@adelaide.edu.au (S.G.);; 3South Australian Health and Medical Research Institute, Adelaide, SA 5000, Australia; 4Garvan Institute of Medical Research, Darlinghurst, NSW 2010, Australia; 5Faculty of Medicine, University of New South Wales, Sydney, NSW 2052, Australia; 6School of Biomedical Sciences, University of New South Wales, Sydney, NSW 2052, Australia; 7School of Clinical Medicine, University of New South Wales, Sydney, NSW 2052, Australia; r.gilchrist@unsw.edu.au

**Keywords:** autofluorescence, spectroscopy, differentiation, stem cells, mesenchymal

## Abstract

Understanding the molecular mechanisms of differentiation is important for regenerative medicine and developmental biology. This study aims to characterise the role of the glycolysis/oxidative phosphorylation balance as a driver of mesenchymal stem cell (MSC) differentiation. Cells were maintained in normal conditions or stimulated towards the MSC trilineage cell types over 21 days. Multispectral imaging of cell autofluorescence was applied as a non-invasive methodology to continuously image cultures in situ. Spectral signals for collagen, NAD(P)H, and flavins were unmixed. MSCs cultured under chondrogenic conditions exhibited increased collagen levels relative to controls. Following osteogenic induction, MSCs showed increased collagen levels relative to controls during the earlier stages of culture; however, control cells increased their collagen levels as they became confluent. MSCs cultured under adipogenic conditions exhibited lower levels of collagen than controls. The redox ratio (RR; NAD(P)H/flavins) immediately decreased during chondrogenesis, with this early effect persisting throughout the culture compared to control cells, which appeared to increase their RR, similar to osteogenesis. Adipogenesis resulted in a small increase in RR on day 2 relative to control cells, followed by a persistent decrease. Chondrogenic and adipogenic differentiation favoured oxidative phosphorylation, whereas osteogenesis and MSC overgrowth resulted in a glycolytic metabolism. Following consideration of these findings, as well as the diverse reports in the literature, it is concluded that neither enhanced oxidative phosphorylation nor glycolysis are fundamental to the canonical modes of differentiation, and researchers should avoid interpreting shifts as indicating differentiation.

## 1. Introduction

Previously, it has been assumed that differentiation and tissue integration are the sole drivers of mesenchymal stem cell (MSC) contribution to regenerative medicine. However, these assumptions have been moderated to include paracrine and autocrine effects [1], as well as emerging recognition of mitochondrial transfer [2,3]. Nonetheless, the study of the molecular mechanisms of MSC trilineage differentiation—osteogenic, adipogenic, chondrogenic—remains a major area of research. Characterising these processes will offer increased insights into tissue maintenance and decline, both of which have the potential to inform our understanding of how to treat related diseases, such as osteoporosis and osteoarthritis. In particular, the general expectation is that stem cells primarily utilise glycolysis for energy generation while undifferentiated but activate oxidative phosphorylation as part of the differentiation process [4]. This metabolic switch has been considered a hallmark of differentiation [5]; however studies have suggested inconsistency, which raises questions as to its importance to the process and relevance as a biomarker [6,7,8].

The assessment of autofluorescence is a common methodology for estimating the metabolic status of cells and tissues in vitro, taking advantage of the endogenous fluorescent properties of select metabolites [9,10]. These include the coenzymes reduced nicotinamide adenine dinucleotide (NAD(P)H) and flavin proteins. NAD(P)H and flavins work together as crucial components of cellular redox reactions and energy transfer processes [11,12]. Their ability to accept and donate electrons makes them central to the efficient functioning of cellular metabolism and ATP production [13]. The relative abundance of NAD(P)H and flavins, based on the intensity of their spectral signatures, is termed the ‘redox ratio’, with increasing amounts of NAD(P)H relative to flavins generally being indicative of increased utilisation of oxidative phosphorylation, while lower amounts of NAD(P)H relative to flavins indicate increased glycolysis [9]. In this report, the redox ratio is formulated as (NAD(P)H/flavins), although different variations (e.g., (flavins/NAD(P)H) or (NAD(P)H/(NAD(P)H+flavins)) can be found in the literature. Care must, therefore, be taken when comparing reports. Collagen, the most abundant protein in the extracellular matrix of connective tissues, is also autofluorescent [14]. Due to the contribution of chondrocytes to cartilage regeneration via collagen production [15], this is particularly relevant to the study of chondrogenesis.

Autofluorophores have identifiable spectral profiles based on their excitation emission characteristics. However, only the most abundant fluorophores can be reliably distinguished from the ‘noise’ of other competing sources of autofluorescence [16]. In this study, we use a spectral microscopy approach, while the differentiation of MSCs is assessed by a broad range of excitation/emission ‘channels’ in order to construct a detailed profile where each pixel represents a full spectral profile which could, subsequently, be unmixed to estimate the concentration of NAD(P)H, flavins, and collagen relative to the total fluorescence. This approach has been successfully applied to the unmixing of autofluorescent signals to identify component fluorophores in several contexts, including the assessment of MSC ageing [17], oocyte quality [18], and the characterisation of cancer cells [19].

An advantage of this technology is that relatively low intensity light can be utilised, which does not damage the continuing viability of the cells [20]. As such, this study aims to investigate the changing profile of collagen, NAD(P)H, and flavins in continuous cultures of MSCs stimulated to undergo osteogenesis, adipogenesis, and chondrogenesis relative to control cells maintained in a normal growth medium over the course of 21 days.

## 2. Materials and Methods

### 2.1. Cell Culture and Differentiation

Human MSCs (hMSCs) were obtained from bone marrow aspirates of healthy adult (aged 18–35) donors from the posterior iliac crest collected and validated as described in [21], including sorting for STRO-1^Bright^/VCAM-1^+^ status. Collection was approved by the Royal Adelaide Hospital Human Ethics Committee under protocol 940911a, while experiments were conducted under the approval of the UNSW low-risk Human Ethics Committee, under protocol HC180219. Two cell lines (ND0142, ND0106) were used in all experiments. All cell cultures were kept in a humidified incubator at 37 °C, 5% CO_2_. Cells were cultured in a normal growth medium (non-differentiated controls), i.e., α-MEM with sodium bicarbonate, without L-glutamine, ribonucleosides, and deoxyribonucleosides (M4526, Sigma, St. Louis, MI, USA), 50 U penicillin 50 mg/mL streptomycin (PS, Sigma, P4333), 10% fetal bovine serum (SH30084, Hyclone, Logan, UT, USA), 5 mM L-glutamine (Gibco, Waltham, MA, USA, 25030081), 100 mM L-ascorbic acid (Sigma, A8960), and 1 mM Sodium pyruvate (Sigma, S8636).

For differentiation, hMSCs were plated at passage 4 in glass bottom coverslip dishes (7.5 × 10^4^ cells per dish). To induce specific cell lineage differentiation, the culture media used were StemPro Chondrogenesis (ThermoFisher, Waltham, MA, USA, A1007101), StemPro Osteogenesis (ThermoFisher, A1007201), and StemPro Adipogenesis (ThermoFisher, A1007001), kit media supplemented with pen and strep for each of the target cell fates. Control cells were maintained in normal hMSC growth medium. All cells were plated and allowed to grow for 24 h in a normal culture media prior to imaging and treatment on ‘day 0’. Media was replaced every 3–4 days, or whenever microscopy was undertaken. On day 21, chondrogenic hMSC cultures were stained with Safranin-O (Sigma, S8884) to assess proteoglycan synthesis (cultures were washed 2× with PBS, fixed in 4% paraformaldehyde (PFA) for 20 min, washed 3× with PBS, stained for 30 min with 0.1% Safranin O in distilled water, washed 3× with PBS). Osteogenic hMSC cultures were stained with alizarin red (Sigma, A5533) to assess calcium deposition (cultures were washed 3× with PBS, fixed in 4% PFA for 20 min, rinsed 3× with distilled water, stained with alizarin red for 30 min (2% adjusted to 4.1 pH to 4.3 pH with HCl), rinsed 5× with distilled water). Adipogenic hMSC cultures were stained with oil red O (Sigma, O0625) to assess lipid formation (diluted 0.5% with oil red O stock with distilled water 3:2, cells washed 2× with PBS, fixed in 4% PFA, washed 2× with distilled water, incubated 5 min in 60% isopropanol, with isopropanol later discarded and the cultures covered with oil red O and then incubated 15 min on the mixer, with the staining product discarded and the cultures washed with water to remove the excess staining liquid). Representative images of stained cultures on day 21 are shown in Figure 1.

### 2.2. Multispectral Microscopy

For imaging, hMSCs were washed twice with the hanks balanced salt solution (HBSS). Spectral microscopy of autofluorescence was conducted using an Olympus IX83 microscope (Tokyo, Japan) with a NuVu electron multiplying charge coupling device camera (hnu1024) with a 40× oil objective lens. Excitation/emission bands were assessed at: 345/414 nm, 345/451 nm, 345/575 nm, 490/575 nm, 505/575 nm, 345/594 nm, 490/594 nm, 505/594 nm, 358/414 nm, 377/414 nm, 381/414 nm, 358/451 nm, 371/451 nm, 377/451 nm, 381/451 nm, 358/575 nm, 371/575 nm, 377/575 nm, 381/575 nm, 391/575 nm, 397/575 nm, 400/575 nm, 403/575 nm, 406/575 nm, 412/575 nm, 418/575 nm, 430/575 nm, 437/575 nm, 457/575 nm, 469/575 nm, 476/575 nm, 358/594 nm, 371/594 nm, 377/594 nm, 457/675 nm, 469/675 nm, and 476/675 nm. Excitation/emission pairs were chosen to include combinations which have previously been optimally informative with respect to unmixing autofluorophore signals [17], balanced against total time taken for imaging, which can be confounded by visible cell migration and reorganisation if excessive (time taken for the imaging protocol was ~5 min per field of view, with variation introduced by manual operation). The technical details of the multispectral microscopy are detailed in [22,23]. Image preparation—the removal of image artifacts, including Poisson’s noise, dead and/or saturated pixels, illumination curvature and background fluorescence—was performed as described in [16,23].

### 2.3. Spectral Unmixing

Linear mixed modelling was applied to compare measured spectral characteristics to the established characteristics of fluorophores and calculate their abundance. The signal of each pixel in the multispectral excitation/emission ‘stack’ was treated as a linear combination of a small number of endmember component spectra whose weights corresponded to the concentration of the responsible autofluorophores. Relative concentrations are expressed as abundance fractions [24,25,26], after the unsupervised algorithm robust dependent component analysis (RoDECA) was used to identify NAD(P)H, flavins, and collagen, all of which were the dominant endogenous fluorophores. RoDECA has been demonstrated to accurately discriminate individual fluorophores despite the presence of overlapping spectra and imaging noise [16,27]. Related fluorophores have similar spectral signatures and, as such, NAD(P)H refers to the combined fluorescence of the fluorophores NADH and NADPH (of which NADH is expected to be the most dominant factor). Flavins is the fluorescence from any flavin family proteins (of which FAD is expected to be the dominant factor) [28,29]. Similarly, collagen fluorescence could originate from any member of the family, although chondrocytes are characterised by the production of type II collagen [15]. Figure 2, Figure 3 and Figure 4 show representative images of parallel ROIs for channels corresponding to excitation/emission peaks of collagen NAD(P)H and flavins for differentiating cells throughout the culture period.

### 2.4. Statistics

Linear mixed modelling statistical assessments were conducted using Matlab (R2017b). Group comparisons were made using the non-parametric Mann–Whitney U test due to violations of the parametric assumptions. Data are presented as cell median values with standard error of the mean (SEM).

## 3. Results

Successful differentiation of osteocytes, chondrocytes, and adipocytes was confirmed by staining with the relevant marker (Figure 1) on day 21. Changes in the autofluorophore relative abundance over time are illustrated in Figure 5 and described in detail in the relevant subsections.

### 3.1. Collagen

Relative abundance of collagen was elevated for MSCs induced under chondrogenic differentiation conditions from 24 h (Figure 6, day 1). Unsurprisingly, this effect persisted through the whole period of culture (Figure 6), becoming increasingly exaggerated from day 7 and continuing until day 21. MSCs induced under osteogenic conditions displayed increased levels of collagen on days 1 to 7 as well as on day 14, but there were no differences in collagen levels on days 7, 11, and 21. Despite initial increases relative to control MSCs (Figure 6, days 1–4), MSCs induced under adipogenic conditions exhibited lower levels of collagen on days 5–14, with a drastic decrease on day 21 (Figure 6). Control MSCs cultured under non-differentiated, normal growth conditions became over confluent, an event associated with loss of pluripotency and spontaneous differentiation. Collagen levels held relatively stable in the early stages of the normal culture, but increased in the later periods, with the highest values observed on day 21 (Figure 5). This followed a similar trajectory to that of MSCs directed towards osteogenic differentiation (Figure 5).

### 3.2. Redox Ratio

The redox ratio (RR) was calculated as NAD(P)H/flavin fluorescence. It was immediately reduced in MSCs induced towards a chondrogenic cell fate, indicating a shift towards greater oxidative phosphorylation in their metabolism. This effect remained strong throughout most of the culture period, with particularly large differences on days 14 and 21 (Figure 7). Of note, non-induced control cells had an elevated RR, having shown steady increases from day 5 (Figure 5). This is indicative of a more glycolytic metabolism, which could result from cellular stress caused by the overconfluence that had been reached at this point. As with collagen, this was similar to the pattern seen for MSCs induced towards osteogenic differentiation. These exhibited small differences from controls on days 2, 3, 5, 7, and 14, but showed no significant effect on days 1, 4, 11, and 21 (Figure 7). On day 21, both groups of MSCs had majorly elevated RRs compared to undifferentiated MSCs on day 0 (Figure 5). MSCs directed towards an adipogenic cell fate experienced a minor increase in RR compared to controls on day 2, and then significant decreases were observed on days 4–21, suggesting a relative shift towards oxidative phosphorylation similar to the chondrocytes.

### 3.3. NAD(P)H

Changes in NAD(P)H appear to have been the main driver of the changes seen in the redox ratio. MSCs directed towards chondrogenesis exhibited significantly lower NAD(P)H relative abundances compared to control MSCs at every timepoint measured. Osteogenic differentiation of MSCs resulted, again, in a similar course to that exhibited by non-induced controls, with no significant differences on day 1–3 and some elevation on days 4–14. Both control cells and osteogenic-induced MSCs appeared to exhibit an increasing relative abundance of NAD(P)H from day 6, but MSCs directed towards osteogenesis maintained a divergence from day 4 (Figure 8) until the occurrence of a steep increase in the controls on day 21 (Figure 8), resulting in loss of significance (Figure 8). MSCs directed towards adipogenesis had higher NAD(P)H levels on day 2, but they then significantly decreased on all subsequent days (Figure 8).

### 3.4. Flavins

Although there were significant differences in flavin levels on all assessed days, effects were comparatively small and relative abundances remained somewhat stable (Figure 9). MSCs cultured under chondrogenic inductive conditions showed significant differences on days 1, 2, 4, 5,11, and 14, with the moderate elevation observed on days 11 and 14 being reversed to a decrease on day 21 due to an increase in control MSCs. Although the relative abundance of flavins was still significantly lower in control MSCs than in MSCs stimulated towards osteogenesis (Figure 9), the gap was considerably narrowed. Adipogenic induction of MSCs caused significant differences in flavin levels compared to controls on days 1, 2, 3, 5, 7, 14, and 21, although, similarly to chondrogenic-induced MSCs, adipogenic-induced MSCs maintained fairly similar levels for the majority of the culture, with the exception of day 21 where the control cell elevation resulted in larger differences (Figure 5).

In Figure 10, cells are plotted in a three-dimensional space according to the relative levels of collagen, NAD(P)H, and flavins in a week-wise manner (i.e., days 7, 14, and 21). Control cells maintained a tight cluster on days 7 and 14; however, a shift occurred on day 21 (of note, the same data are plotted for control cells in all three graphs to allow for comparison, with the angle of view chosen to best visualise the effects on the differentiation of MSCs). Chondrogenic-induced MSCs appeared to diverge away from control cells on days 7 and 14, while maintaining their own tight cluster. On day 21, however, the chondrogenic-induced MSCs appeared to have greater heterogeneity, although there was still a visibly distinct separation from the control MSCs. For osteogenesis, the differentiating MSCs maintained a tight cluster at all assessed timepoints, with the control cells appearing to migrate towards this cluster on day 21. Finally, with respect to adipogenesis, control and differentiating cells maintained tight, overlapping clusters on days 7 and 14, although a shift was observed for cells undergoing adipogenesis on day 14, which became more strongly noticeable on day 21. This was accompanied by a loosening of cluster cohesion.

## 4. Discussion

Understanding the molecular mechanisms of cell fate determination is important for regenerative medicine and developmental biology. This study goes beyond prior works in the area [5,6,7,8,30,31,32] by covering the full range of MSC trilineage differentiation potential and by unmixing signals for NAD(P)H, flavins, and collagen. An additional point of novelty is that, owing to the non-invasive nature and low intensities of the light applied in our multispectral approach to the imaging of autofluorescence, we were able to undertake the longitudinal assessment of continuous cultures rather than assessing parallel cultures at different timepoints. Consideration of spectral profiles of differentiating cells for the three autofluorophores (Figure 10) over time suggest increasing heterogeneity despite the maintenance of visible groups.

In previous work, we have shown that the ratio of NAD(P)H to flavins in MSCs decreases with increasing cell age [17], indicating a metabolic shift towards oxidative phosphorylation compared to glycolysis. Increasing age in MSCs has several biological consequences, including the loss of osteogenic differentiation capacity in favour of adipogenesis. As such, our finding that osteogenesis is associated with elevated NAD(P)H levels and an increased ratio of NAD(P)H to flavins (suggesting increased glycolysis), while adipogenesis is associated with a falling redox ratio and lower NAD(P)H levels, is consistent with this pattern. As well as stimulating differentiation and concomitant metabolic changes, the composition of the media used has the potential to directly impact the autofluorophore abundance in the cells. Specifically, the adipogenic differentiation conditions included high glucose, whereas, in the other environments, glucose levels were low. Although the exact effect of this difference cannot be determined, if it was major, it would be expected that the metabolic state of the MSCs in the adipogenic conditions would diverge at the earliest timepoint. However, in Figure 5, the redox ratio is similar among undifferentiated, osteogenic, and adipogenic cells.

Numerous studies have been undertaken on the metabolomics of human-bone-marrow-derived MSC differentiation, with a particular emphasis on osteogenic differentiation [5,7,8,30,31] and with generally divergent results. Our findings have the greatest agreement with those by Rice et al., who used two-photon-excited fluorescence microscopy to investigate osteogenesis, adipogenesis, and proliferating MSCs (analogous to our control treatment) [33]. As in the present report, they showed that, during osteogenesis, both NAD(P)H and flavin concentrations generally increased over the course of 21 days. Additionally, they also found that MSCs which were not directed to differentiate towards a specific cell fate experienced a similar effect. In their work, collagen deposition, assessed by second harmonic generation, was generally increased in MSCs undergoing osteogenesis relative to undifferentiated cells. We saw little difference between control and osteogenic cells at most timepoints, with no significant effect on day 21. However, Figure 5, which illustrates the trend over time in a way that better matches the statistical comparisons made by Rice, suggests concordant results. Under their system, no collagen was detected at any timepoint for cells undergoing adipogenesis, whereas we found a detectable signal which was consistently lower in MSCs undergoing adipogenesis compared to control cells after the earlier stages of differentiation (day 5–21, Figure 5).

In contrast, Guo et al. reported increased fluorescence lifetimes for NADH (generally associated with lower concentrations) during osteogenesis in two separate studies [30,31]. In the more recent publication, they also demonstrated a shift towards oxidative phosphorylation, which is also typified by lower levels of NADH relative to flavins [31]. Conversely, in 2017, Meleshina et al. reported that differentiation of MSCs to osteogenic cell fates is accompanied by a shift towards a more glycolytic metabolism—after an initial period of increased oxidative phosphorylation—as indicated by an increasing ratio of NAD(P)H to FAD (reported in their manuscript as a decreasing redox ratio under the FAD/NAD(P)H formulation) [5]. This agrees with our findings. However, in another study, MSCs undergoing osteogenesis were shown to have a stable rate of glycolysis accompanied by increasing oxidative phosphorylation [8]. These and our findings were all achieved through in vitro observation, and it should be noted that the in vivo environment could greatly influence the characteristics of MSC differentiation.

Fewer studies have been undertaken on chondrogenesis and adipogenesis. In a separate work, Meleshina et al. found that adipogenesis increased NAD(P)H concentrations relative to FAD from day 19 after an extended period of no effect [6], whereas we found declines relative to undifferentiated controls from day 4. The actual pattern of the effect relative to fully undifferentiated cells on day 0 was comparatively inconsistent, but the redox ratio on day 21 was lower than that on day 0 (despite being similar on the previous day 14 timepoint). Rice et al. also found increased NAD(P)H levels with adipogenesis [33], although such a phenomenon became observable from the earliest stages of adipogenesis rather than later in the process. As this was accompanied by generally increasing flavin levels, however, they saw no clear effect on redox ratio over time.

MSCs directed towards chondrogenic differentiation by pellet culture have been shown to exhibit reduced oxygen consumption, indicating a shift towards a glycolytic metabolism [7]. This shift was also observed in two-dimensional cultures based on the ratio of NAD(P)H and FAD lifetimes [5]. However, our observation of a falling ratio of NAD(P)H to flavins relative to undifferentiated and control cells (Figure 5 and Figure 7) indicates that there was a shift towards oxidative phosphorylation during chondrogenesis in our system. Our observation of increasing collagen during chondrogenic differentiation, however, conforms with the findings of [5].

## 5. Conclusions

MSC lines are known to differ considerably in their characteristics based on derivation, culture environment, and prior exposures. With consideration of how the correlations of the assessed autofluorophores become more variable over time—for both differentiating and control MSCs (Figure 10)—as well as of diverse reports for metabolic profiles for differentiated MSCs in the broader literature, it must be concluded that neither enhanced oxidative phosphorylation nor glycolysis are fundamental to the canonical modes of differentiation, which, in vitro, have been shown to occur successfully in the presence of different redox balances. Researchers should avoid interpreting shifts from glycolysis to oxidative phosphorylation (and vice versa) as indicators of differentiation.

## Figures and Tables

**Figure 1 cells-13-01731-f001:**
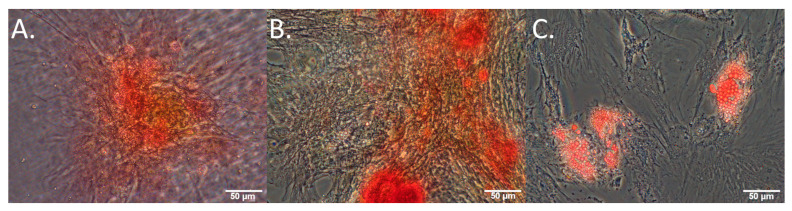
Cultured MSCs stimulated towards (**A**) chondrogenesis, (**B**) osteogenesis, (**C**) adipogenesis, all stained with safranin-O, alizarin red, and oil red O, respectively, on day 21 of culture.

**Figure 2 cells-13-01731-f002:**
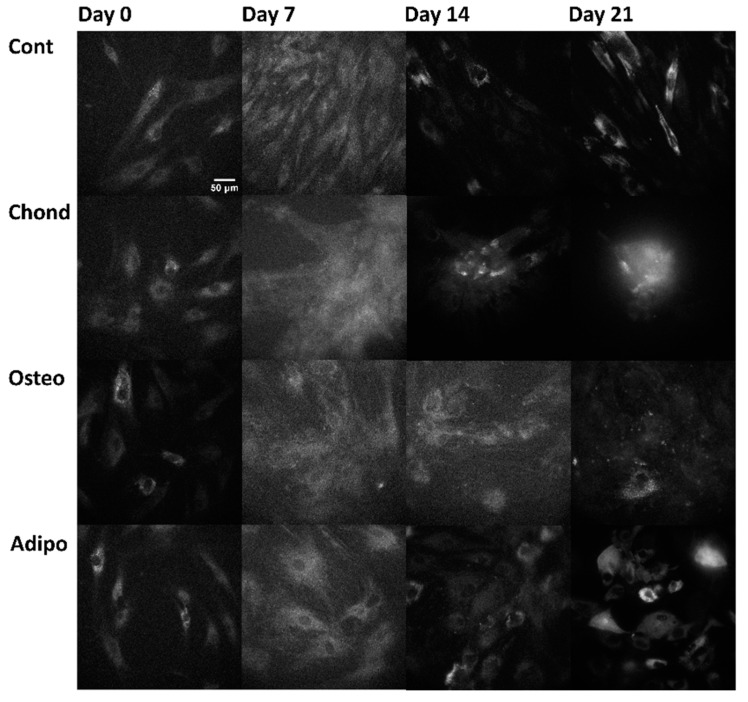
Collagen (not specific to any type) autofluorescence (arising from the excitation at 371 nm and emission at 414 nm, channel where collagen is the dominant—although not the sole—autofluorophore) from control, chondrogenic, osteogenic, and adipogenic cells on days 0, 7, 14, and 21.

**Figure 3 cells-13-01731-f003:**
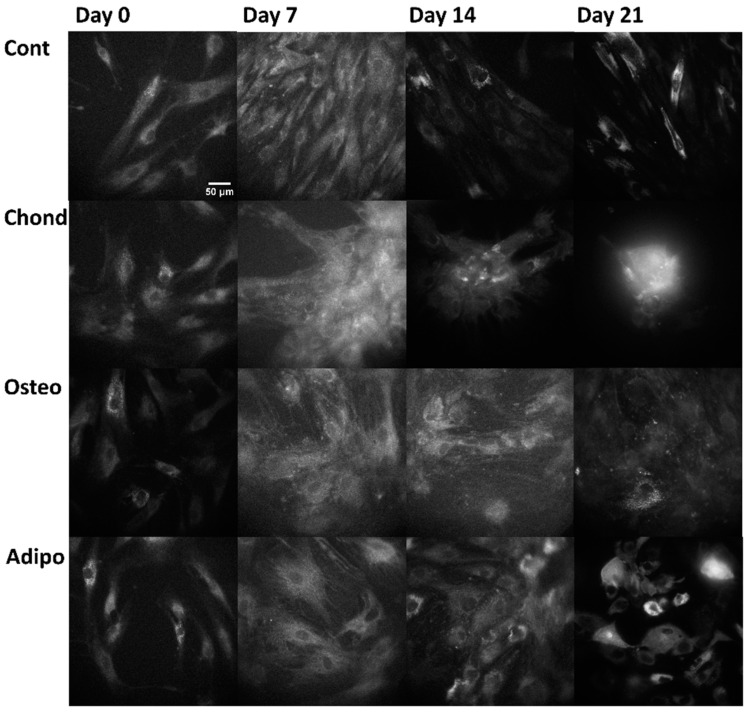
NAD(P)H autofluorescence, arising from the excitation at 345 nm and emission at 451 nm (where NAD(P)H is the dominant—although not the sole—autofluorophore), from control, chondrogenic, osteogenic, and adipogenic cells on days 0, 7, 14, and 21.

**Figure 4 cells-13-01731-f004:**
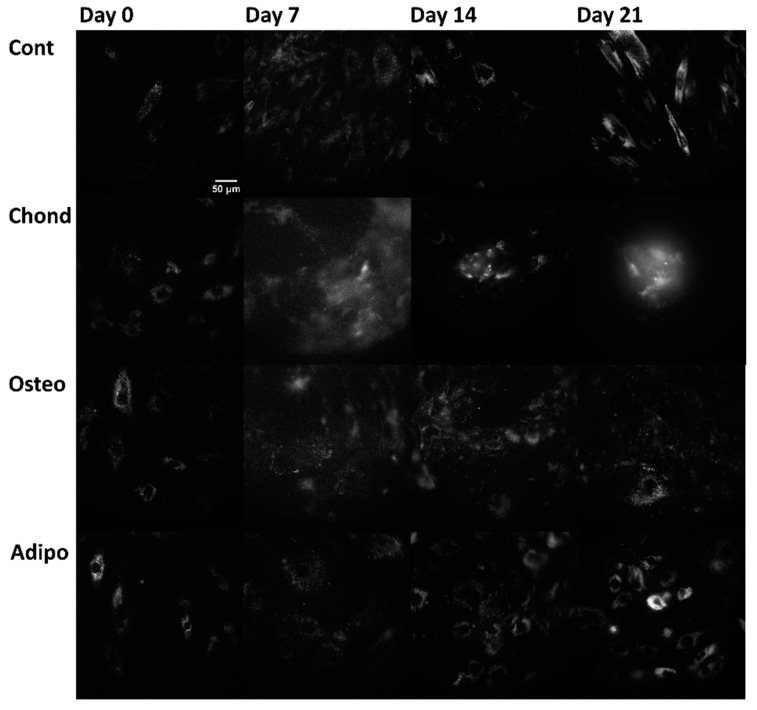
Flavin autofluorescence, arising from the excitation at 490 nm and emission at 575 nm (where flavin proteins are the dominant—although not the sole—autofluorophores), from control, chondrogenic, osteogenic, and adipogenic cells on days 0, 7, 14, and 21.

**Figure 5 cells-13-01731-f005:**
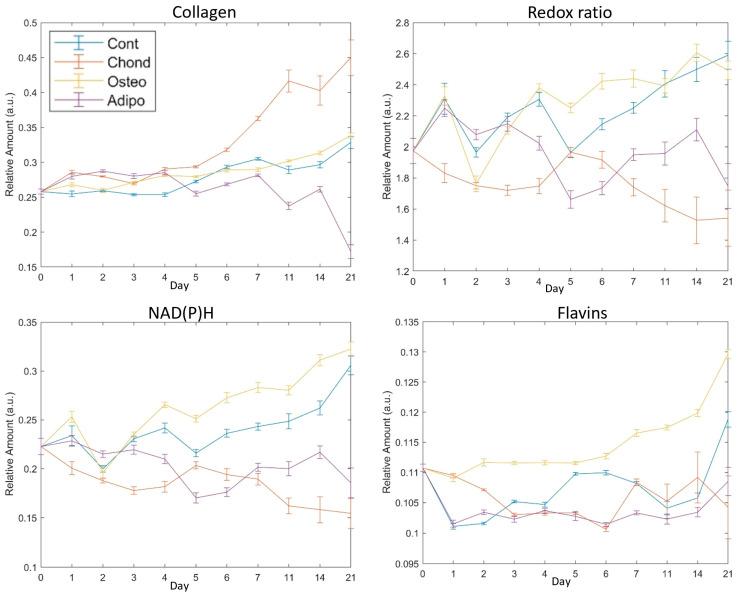
Mean values of fluorophores with SEM error bars on each assessed day for collagen and the redox ratio of NAD(P)H to flavins. Day 0 indicates cells with no exposure to the differentiation media, day 21 was the final assessment day. The redox ratio is NAD(P)H/flavins.

**Figure 6 cells-13-01731-f006:**
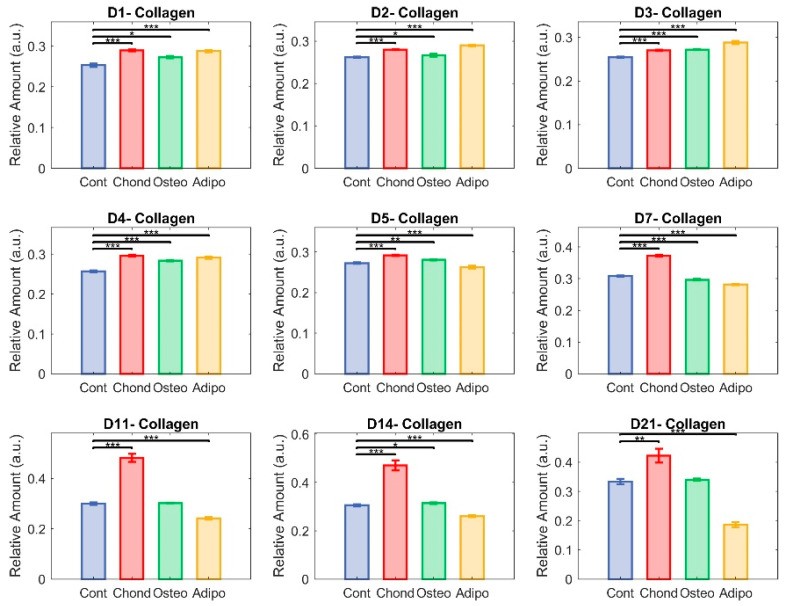
Unmixed collagen. Here, each group—control (in blue), chondrocytes (in red), osteocytes (in green), and adipocytes (in yellow)—is represented with respect to their relative content of collagen. The non-parametric Mann–Whitney U test was used to assess significance (* *p* < 0.05, ** *p* < 0.01, and *** *p* < 0.001) among groups.

**Figure 7 cells-13-01731-f007:**
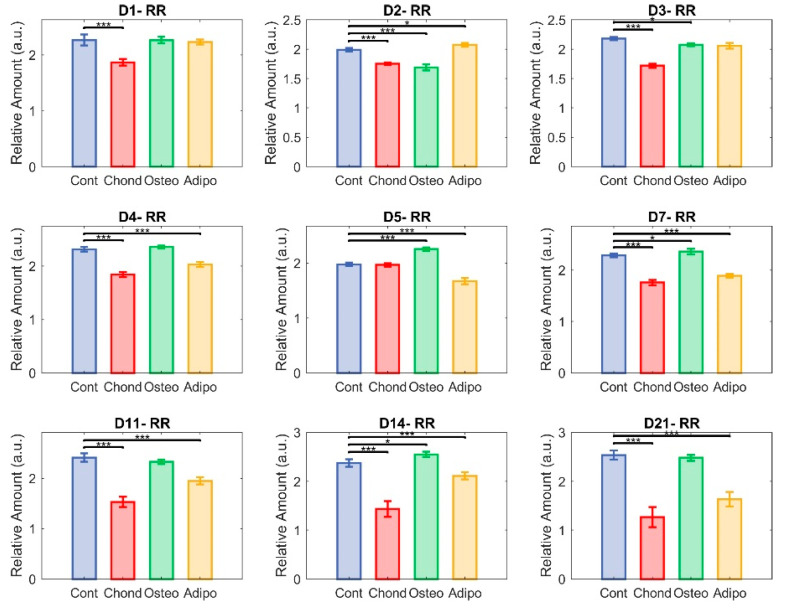
Redox ratio (RR). Each group—control (in blue), chondrocytes (in red), osteocytes (in green), and adipocytes (in yellow)—represent NAD(P)H relative abundance divided by flavin relative abundance in the cultured cells (* *p* < 0.05 and *** *p* < 0.001).

**Figure 8 cells-13-01731-f008:**
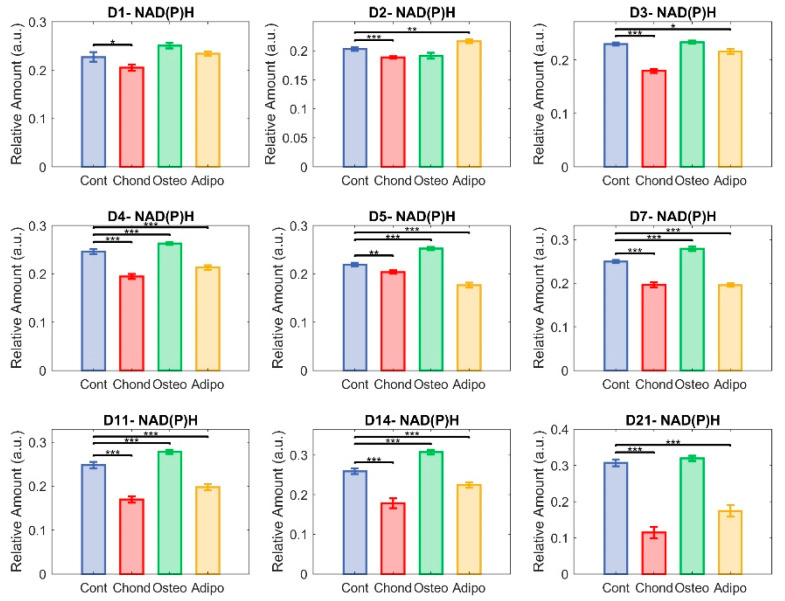
Unmixed NAD(P)H. Here, each group—control (in blue), chondrocytes (in red), osteocytes (in green), and adipocytes (in yellow)—is represented in terms of the relative abundance of NAD(P)H (* *p* < 0.05, ** *p* < 0.01, and *** *p* < 0.001).

**Figure 9 cells-13-01731-f009:**
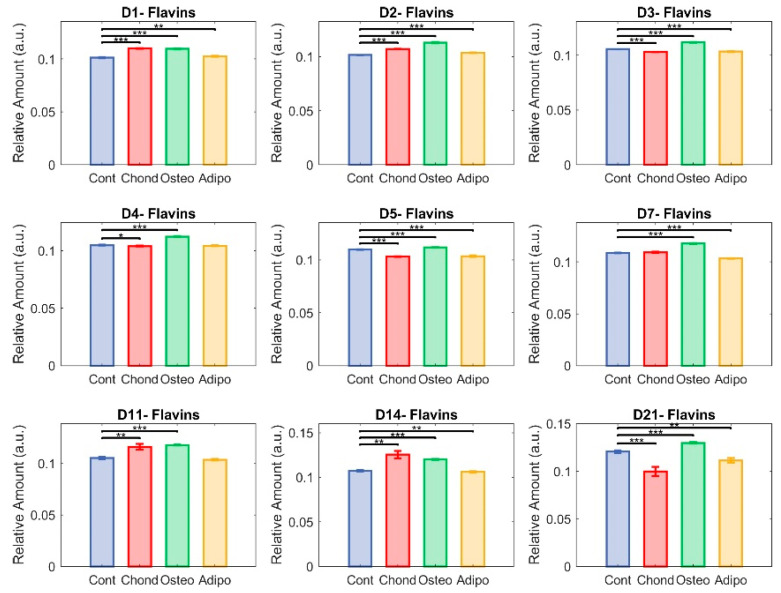
Unmixed flavins. Here, each group—control (in blue), chondrocytes (in red), osteocytes (in green), and adipocytes (in yellow)—is represented in terms of relative abundance of flavins (* *p* < 0.05, ** *p* < 0.01, and *** *p* < 0.001).

**Figure 10 cells-13-01731-f010:**
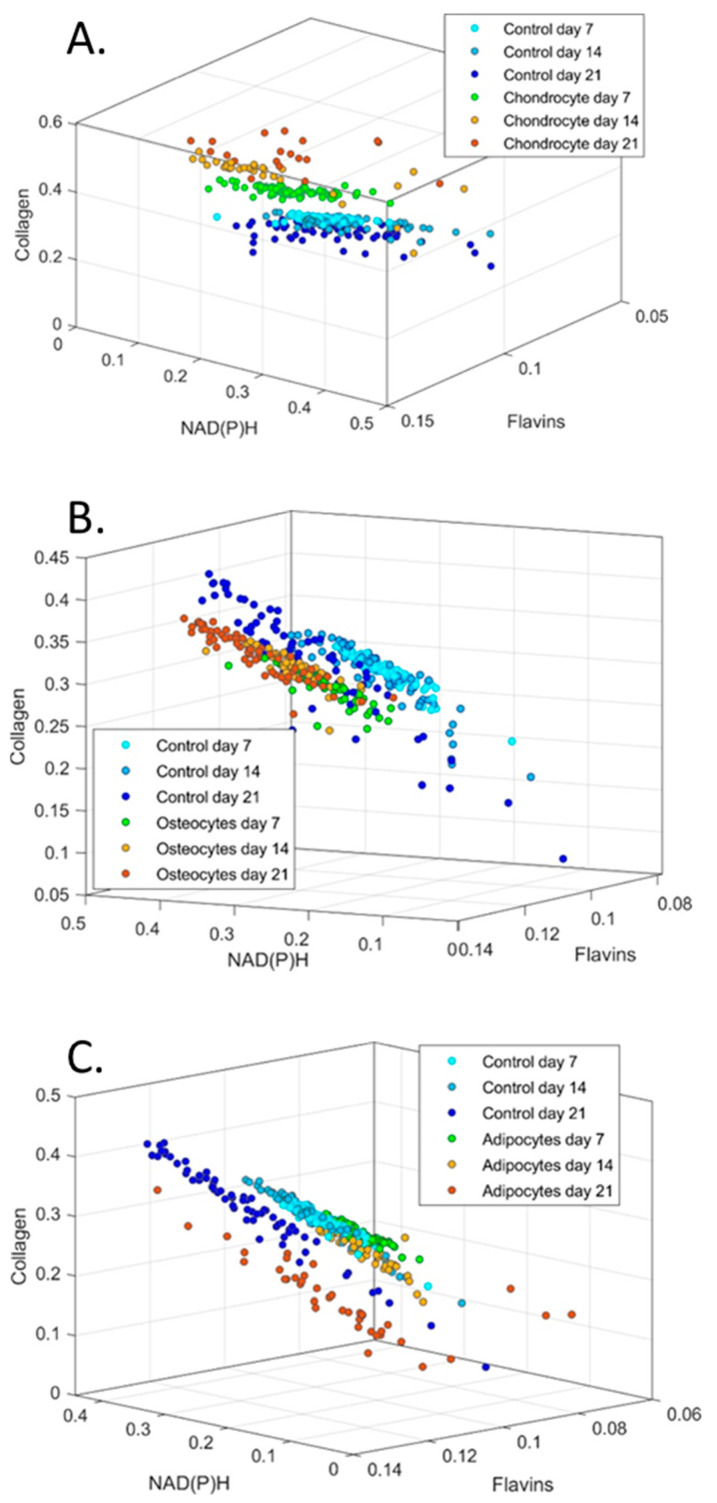
Cluster graphs of relative abundance of collagen, NAD(P)H and flavins plotted against each other for (**A**) chondrocytes, (**B**) osteocytes and (**C**) adipocytes. 3D graphs have been oriented to best show groupings. Control cells are plotted in each group for comparison.

## Data Availability

Data will be made available to qualified researchers upon reasonable request.

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
