# Peer review of "Multispectral Imaging of Collagen, NAD(P)H and Flavin Autofluorescence in Mesenchymal Stem Cells Undergoing Trilineage Differentiation"

_cells, 2024, doi:10.3390/cells13201731_

Round 1
Reviewer 1 Report
Comments and Suggestions for Authors
The manuscript shows the difference in aerobe or anaerobe turnover in adiopgenic, osteopenia and chondrogenic differentiation of human MSC by elegant spectral microscopy. The results are helpful in understanding basic mechanisms of stem cell differentiation.
There are only minor remarks:
The source and characteristics of the human MSCs applied, and how many they used, should be given.
Author Response
The source and characteristics of the human MSCs applied, and how many they used, should be given.
These additional details are now given in the Materials and Methods section (2.1 Cell culture and differentiation), as follows:
Human MSC (hMSC) were obtained from bone marrow aspirates of healthy adult (aged 18-35) donors from the posterior iliac crest collected and validated as described in [21], including sorting for STRO-1Bright/VCAM-1+ status . Collection was approved by the Royal Adelaide Hospital Human Ethics Committee under protocol 940911a, while ex-periments were carried out under the approval of UNSW low-risk Human Ethics Com-mittee under protocol HC180219. Two cell lines (ND0142, ND0106) were used in all ex-periments.
Reviewer 2 Report
Comments and Suggestions for Authors
The proposed manuscript reveals a fascinating, noninvasive approach to revealing the role of the balance between glycolysis and oxidative phosphorylation as a driver of mesenchymal stem cell (MSC) differentiation.
The methodology is described in very detail and could be easily reproducible.
General questions:
1. Since the differentiation media for osteogenesis, chondrogenesis, and adipogenesis) contain (or not) different glucose concentrations, how did you subtract this effect from the observed changes?
2. In fact, all the controls are the same cells since they were treated with non-differentiating media. To trace if there are significant changes during the whole period, I would suggest an additional statistical analysis to be performed where the differences between inner groups ( for example, in chondro group between D1, D7, D14 and D21) for Unmixed collagen, Redox ratio (RR), NAD(P)H and flavins in a comparative manner. Those data could be explained only in one sentence for each parameter.
Technical corrections:
Line 361: At the end of the sentence is a missing dot.
On the first page, the type of article is mentioned as a review. Shouldn't it be an Article?
Author Response
General questions:
- Since the differentiation media for osteogenesis, chondrogenesis, and adipogenesis) contain (or not) different glucose concentrations, how did you subtract this effect from the observed changes?
The potential for the direct effect of exposure to different factors in culture media to be different from the consequences of that exposure (i.e. differentiation) is an interesting point. In our system the osteogenic and chondrogenic differentiation environments were both low glucose, while the adipogenic environment was high. The undifferentiated medium was also a low glucose environment. Unfortunately the impact of medium versus differentiation cannot be strictly de-lineated. However if glucose concentration were having a major effect on the metabolic state (and resultant autofluoresence) of the cells, there would be a rapid diversion in the characteristics of cells exposed to the high glucose medium (adipose cells) and all other cell types. However, consideration of the redox ratio over time (Fig. 5) illustrates that this effect does not occur, with adipogeneic cells having a very tight overlap with control and osteogenic cells at day 1. We now detail our consideration of this point in the discussion as follows:
“As well as stimulating differentiation and concomitant metabolic changes the composi-tion of the media used has the potential to impact the autofluorophore abundance of cells directly. Specifically, the adipogenic differentiation conditions were high glucose, where as the other environments were low. Although the exact effect of this difference cannot be delineated, if it was major it would be expected that the metabolic state of the MSC in the adipogenic conditions would diverge at the earliest point, however in Fig. 5 redox ratio is closely similar for undifferentiated, osteogenic and adipogenic cells.”
- In fact, all the controls are the same cells since they were treated with non-differentiating media. To trace if there are significant changes during the whole period, I would suggest an additional statistical analysis to be performed where the differences between inner groups ( for example, in chondro group between D1, D7, D14 and D21) for Unmixed collagen, Redox ratio (RR), NAD(P)H and flavins in a comparative manner. Those data could be explained only in one sentence for each parameter.
We have considered making these comparisons in our initial submission, but made the decision to focus on the treatment-to-control comparisons as it gave the most reliably interpretable results (a single sentence description as you describe would be desirable, however the periodic reversals of metabolite levels (e.g. adipogenic rr sinks to day 5 decreased to day 5 before increasing to day 14) would make such a synthesis considerably longer and very difficult for readers to interpret. We believe that Figure 5, which illustrates changes over time along with SEM error bars provides the optimum solution to the challenge of clearly communicating this information to the reader.
Technical corrections:
Line 361: At the end of the sentence is a missing dot.
This is now corrected.
On the first page, the type of article is mentioned as a review. Shouldn't it be an Article?
Yes. This is now corrected.
Reviewer 3 Report
Comments and Suggestions for Authors
The authors reported a novel method for characterizing the role of the glycolysis/oxidative phosphorylation balance as a driver of MSC differentiation. Such method is able to realize MSC differentiation analysis in a non-damage manner. It is suggested to be accepted after addressing the follow minor concerns:
1. It should be a concluding sentence in the end of Abstract to address the main novelty (and further possible downstream applications if applicable) of this work.
2. Figure1: The font under the scale bar in this figure should be enlarged.
3. Figure 5: It seems a little strange to put a single figure ahead of section 3.1. Maybe the contents related to Figure 5 could become a new section. And it is suggested to illustrate the curves in Figure 5 in details. Also, the words inside in this figure are not clear enough, the authors should improve the figure resolution.
4. Figure 10: It is suggested to have a new section to illustrate relative abundance of collagen, NAD(P)H and flavins.
5. In Discussion, the authors argue that their results are consistent with the Ref[5]. But both works have used 2D cell cultures, since MSCs growing environment in vivo is typically a 3D environment, will such results altered if culturing MSCs in 3D environment?
Author Response
- It should be a concluding sentence in the end of Abstract to address the main novelty (and further possible downstream applications if applicable) of this work.
The following conclusion has been added to the abstract:
In consideration of these findings, as well as diverse reports in the literature, it is concluded that neither enhanced oxidative phosphorylation nor glycolysis are fundamental to the canonical modes of differentiation, and researchers should avoid interpreting shifts as indicating differentiation.
- Figure1: The font under the scale bar in this figure should be enlarged.
Font size has been increased.
- Figure 5: It seems a little strange to put a single figure ahead of section 3.1. Maybe the contents related to Figure 5 could become a new section. And it is suggested to illustrate the curves in Figure 5 in details. Also, the words inside in this figure are not clear enough, the authors should improve the figure resolution.
We have placed the text of the introductory statement above Figure 5 and added some further intext explanation of what this figure conveys. However we do not believe that it is abnormal practice to have an introductory section such as this above the first subheading. Greater explanation has been added to the figure 5 legend to better illustrate the curves. A larger version of the text box with higher resolution text has been added.
- Figure 10: It is suggested to have a new section to illustrate relative abundance of collagen, NAD(P)H and flavins.
Relative abundance of NAD(P)H and flavins is explored in section 3.2 on redox ratio as this is a key indicator of cellular metabolomics. We do not believe that consideration of the abundance of the structural protein collagen relative to these metabolites will add any insights to the manuscript beyond those already provided by the 3D graph of figure 10 which illustrates the abundance of each autofluorophore relative to the others in each cell via position.
- In Discussion, the authors argue that their results are consistent with the Ref[5]. But both works have used 2D cell cultures, since MSCs growing environment in vivo is typically a 3D environment, will such results altered if culturing MSCs in 3D environment?
It cannot be extrapolated from our findings how 3D culture may alter the metabolism of differentiating MSCs. However we now note this consideration in the Discussion as follows:
“These and our findings were all made through in vitro observation, and it should be noted that the in vivo environment could greatly influence the characteristics of MSC differentia-tion.”